# Event-Focused Digital Control to Keep High Efficiency in a Wide Power Range in a SiC-Based Synchronous DC/DC Boost Converter

**María R. Rogina** *[ID], **Alberto Rodríguez** [ID], **Aitor Vázquez** [ID], **Diego G. Lamar** [ID] and **Marta M. Hernando** [ID]

Electronic Power Supply Systems Group, University of Oviedo, Campus de Viesques, 33204 Gijon, Spain; rodriguezalberto@uniovi.es (A.R.); vazquezaitor@uniovi.es (A.V.); gonzalezdiego@uniovi.es (D.G.L.); mmhernando@uniovi.es (M.M.H.)
* Correspondence: rodriguezrmaria@uniovi.es

**Abstract:** This paper is focused on the design of a control approach, based on the detection of events and changing between two different conduction modes, to reach high efficiency over the entire power range, especially at medium and low power levels. Although the proposed control strategy can be generalized for different topologies and specifications, in this paper, the strategy is validated in a SiC-based synchronous boost DC/DC converter rated for 400 V to 800 V and 10 kW. Evaluation of the power losses and current waveforms of the converter for different conduction modes and loads predicts suitable performance of quasi-square wave mode with zero voltage switching (QSW-ZVS) conduction mode for low and medium power and of continuous conduction Mode with hard switching (CCM-HS) for high power. Consequently, this paper proposes a control strategy, taking advantage of digital control, that allows automatic adjustment of the conduction mode to optimize the performance for different power ranges.

**Keywords:** conduction mode change; event-focused control; high efficiency at light load; QSW; SiC bidirectional boost

## 1. Introduction

Energy storage systems, renewable energies, energy recovery systems, power electronic transformers, DC distribution grids, and smart grids [1–6] are some examples of current topics related to power electronics. Most of these applications require bidirectional dc–dc power converters and energy storage systems. Together with that, the battery charging/discharging process is usually executed in three stages [7], with a final stage in which the charging current is very low. Therefore, power converters working as an interface between the storage system and the rest of the grid must achieve high efficiency over a wide power range.

An interesting topic related to these applications is the integration of distributed energy resources in multilevel converters [8]. By means of the adequate design of the cells of multilevel converters (with a cell voltage usually around 1 kV), it is possible to integrate low voltage DC or AC power sources (such as wind turbines or photovoltaic panels), loads or energy storage systems at the cell level [9–11].

Hence, a power converter designed to integrate batteries at the cell level in a multilevel converter must withstand high voltage, while at the same time providing high efficiency (Figure 1). The use of wide band gap (WBG) semiconductors, especially silicon carbide (SiC) MOSFETs, allows power converters to operate at high voltage and high switching frequency, keeping high efficiency [12]. SiC MOSFETs and a variable switching frequency control technique providing zero voltage switching (ZVS) have been used to improve efficiency in a synchronous boost converter while taking into account

the current ripple stress (which may be damaging during the battery charging process [7]), especially at medium and low load operating at high voltage and high frequency [13,14].

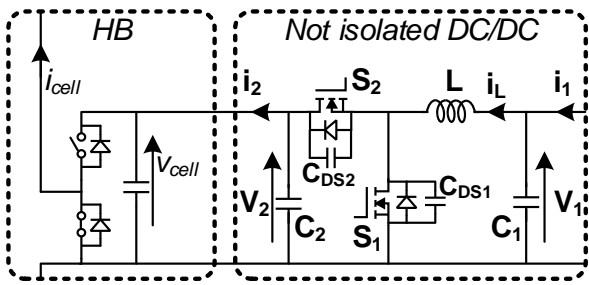

**Figure 1.** Basic cell structure of a multilevel converter integrating energy resources. This cell is composed of a Half-Bridge (HB) cell and synchronous boost DC/DC converter connected in cascade [13].

These days, most of the high-power converters developed are digitally controlled, providing high versatility, for example, making possible automatic correction of the conduction mode. In this paper, a control strategy, based on theoretical efficiency models, that changes the conduction mode for different power levels is presented. The idea is to achieve the maximum possible performance of a SiC-based synchronous boost DC/DC converter. Despite the fact that this manuscript is oriented to provide energy storage capability in a multilevel converter by developing a bidirectional boost converter, the extracted conclusions can also be applied to other applications and converter topologies where high efficiency in a wide power range is needed (some of the applications of interest would be wind energy generation with storage capability [11] or electric vehicle battery chargers [15,16]). Also, the control solution proposed, and the conclusions may be extended to converters using a different semiconductor technology.

This paper is organized as follows. Section 2 outlines the main characteristics of the different conduction modes used in the implementation of the control strategy. In Section 3, the main contribution of this work is presented. An extensive description of the implementation of the digital control strategy is presented, including simulation models and results. Moreover, this section highlights the design of the voltage regulator, the procedure followed to change between conduction modes and the event-focused method followed to achieve quasi-square wave mode with zero voltage switching (QSW-ZVS) automatically. In Section 4 the most relevant experimental results obtained to validate the control strategy proposal are shown. Finally, conclusions are addressed in Section 5.

## 2. Outline of the Conduction Modes under Study

Different conduction modes have been suggested and analyzed for synchronous boost DC/DC converter. Figure 2 shows the key waveforms that define the two selected conduction modes used in our control strategy (two types of continuous conduction mode, CCM). Although more conduction modes have been analyzed, following [14] conclusions, these two modes provide the best performance in terms of efficiency over a wide power range for the proposed application.

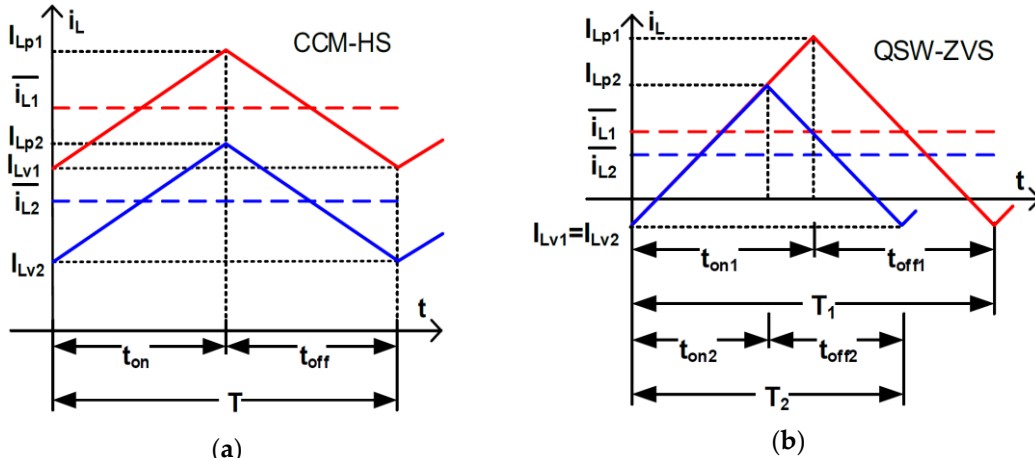

**Figure 2.** Ref. [13] Inductor current waveform for two different power levels. (**a**) CCM-HS and (**b**) QSW-ZVS.

(1)　CCM hard switching (CCM-HS) in Figure 2a. Its main advantages are constant switching frequency (f), low conduction losses, and low current ripple (appropriate for charging and discharging energy storage systems). In contrast, high-switching losses constitute the main disadvantage. At light loads, this mode can achieve ZVS (i.e., triangular current mode (TCM) at constant switching frequency).

(2)　QSW-ZVS [17] in Figure 2b. Large current ripple (the inductance current is negative at the turn-on of $S_1$) and variable switching frequency comprise its main characteristics. ZVS can be achieved by suitable selection of the minimum value of the inductance current and dead-time for different input and output voltage relationships [18], thereby reducing switching power losses. The main drawback is its high conduction losses due to the large current ripple.

Analytical models for the efficiency evaluation of a bidirectional SiC-based boost converter working under the previously described conduction modes are developed and experimentally verified (Figure 3) [19]. The main specifications are: $V_1$ = 400 V, $V_2$ = 800 V, maximum power of 10 kW, and switching frequency of 60 kHz for CCM-HS and from 20 kHz to 200 kHz for QSW-ZVS (full specifications of the converter are given in Section 4).

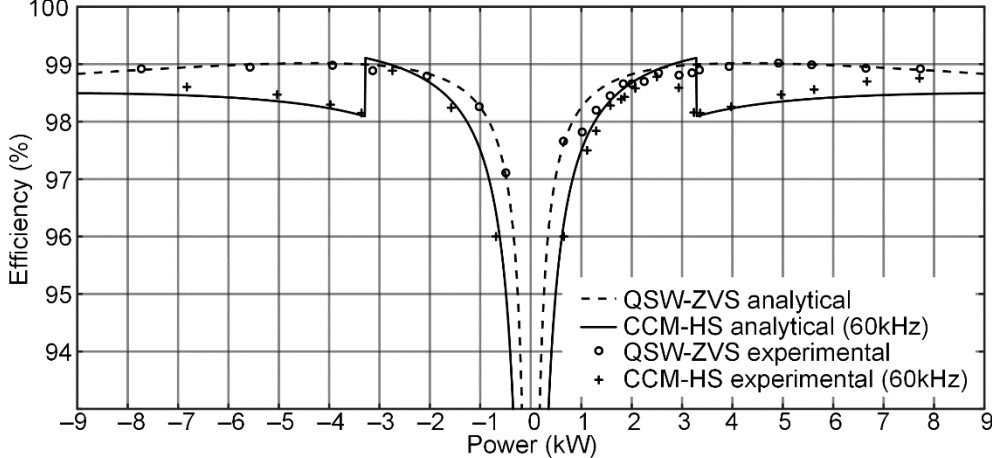

**Figure 3.** Analytical and experimental efficiency comparison. Positive power is considered when the power flows from $V_1$ to $V_2$. And, conversely, negative power represents power flowing from $V_2$ to $V_1$.

Among the different technologies of commercially available semiconductors, SiC MOSFETs are selected since high efficiency operation of the power converter even under high switching frequencies and high voltage requirements is reachable [12]. SiC MOSFETs allow an extensive variation of the switching frequency, which is especially important at low power, where high frequency is needed to maintain QSW-ZVS while achieving high efficiency.

Using SiC MOSFETs and a variable switching frequency control strategy, high performance in a synchronous DC/DC boost converter is obtained. This procedure is favorable to provide ZVS especially at light and medium load operating conditions. Nevertheless, the main drawback of this conduction mode is its high current ripple, which affects directly to conduction losses, especially at full load [18,19]. To make good use of the different conduction modes and to obtain high performance of the converter for the whole power range, a digital control solution changing between conduction modes is presented.

## 3. Implementation of the Digital Control Strategy and Simulation Results

The implementation of the proposed control strategy has been done using different blocks with specific tasks. The blocks diagram of the control system proposed is presented in Figure 4. The control system, including all the presented blocks, is simulated using Matlab-Simulink® and implemented in an FPGA to control the boost converter prototype. The principle of operation of the control strategy, which was introduced in [20], is now presented in more detail in this paper, and extended to the bidirectional operation.

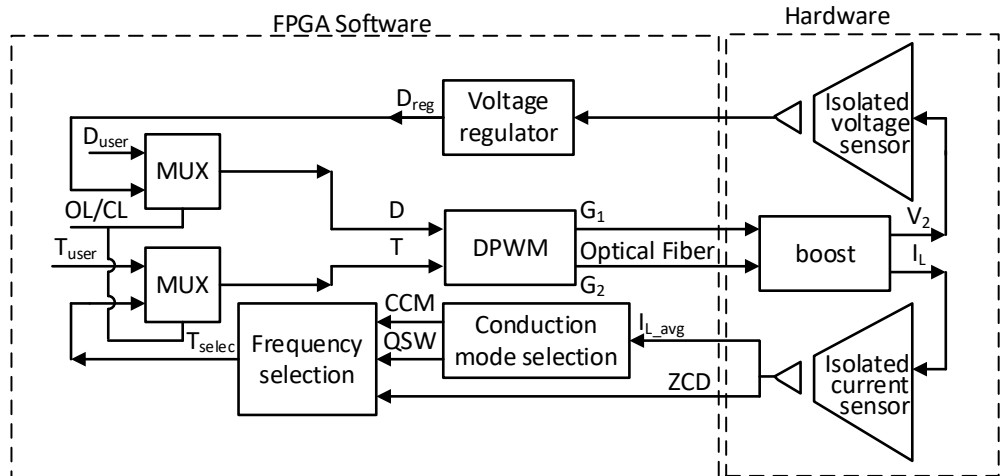

**Figure 4.** Schematic of the control system implemented.

The boost converter block of Figure 4 is the synchronous DC/DC boost converter, further described in Section 4. This block includes the SiC power stage and the drivers. Gate signals ($G_1$, $G_2$) are the input signals of the driver of the boost block, which are isolated using means of optic fiber. At this point, the synchronous boost DC/DC converter is ready to process a specific power changing the voltage of the port 1 ($V_1$, Figure 1) in another voltage at the port 2 ($V_2$), or vice versa. The proposed control strategy takes as the outputs of this block the two variables required, $V_2$ and the inductor current ($I_L$).

The remainder blocks are logically implemented in an FPGA, except for the isolated current and voltage sensors (which are described in Section 4, too).

The aim of the DPWM block is to generate the accurate gate signals required by the driver of the synchronous boost DC/DC converter. These gate signals are generated varying the duty cycle (D) and/or the period (T) depending the conduction mode selected. This decision is taken in each switching period. DPWM block also sets the mandatory deadtimes needed to avoid any undesired behavior, such as short-circuits between devices.

MUX blocks are implemented to assure full control of the system during the failure procedure test. When the synchronous boost DC/DC converter is working in Closed Loop (CL), MUX blocks provide $G_1$ and $G_2$ with the duty cycle and period calculated by the voltage regulator and the frequency selection block ($D_{reg}$ and $T_{selec}$, respectively). Different from that, during failure procedure tests, Open Loop (OL) operation may be chosen, and any wanted value ($D_{user}$ and $T_{user}$) may be defined in order to test the synchronous boost DC/DC converter working at any operating condition.

The blocks voltage regulator, conduction mode selection and frequency selection are further detailed through the next sections. They are the key blocks of the event-based control strategy under development. The main objectives of these blocks are: regulating the voltage $V_2$, selecting the conduction mode to maintain high performance of the boost converter and selecting the switching frequency (constant switching frequency for CCM-HS and variable switching frequency for QSW-ZVS).

It is important to comment that the communication between the synchronous DC/DC converter (boost block) and the digital control stage (rest of blocks) is done through optic fiber, isolating both stages and therefore providing more reliable working conditions of the system.

### 3.1. Voltage Regulator

The proposed control strategy is validated in a bidirectional boost converter, being $V_1$ fixed by a voltage source and $V_2$ regulated by the own converter. The dynamic requirements of the voltage feedback loop are generally specified by the application, but they are not relevant to the validation of the proposed control strategy. Consequently, the design of this feedback loop is not the contribution of this paper, but it is needed. A digital PI voltage regulator that assures the stability of the system in the worst case is designed, obtaining a stable response for both conduction modes (QSW-ZVS and CCM-HS) [21]. A smooth and slow regulation is looked for, to avoid damages of the system.

In Figure 5, the simulated response of the designed voltage regulator with a step in $V_1$ is shown. As can be seen, an overdamped response is obtained (similar experimental results are obtained and presented in Section 4). The PI regulator has been further tested under steps in the reference voltage and load steps fitting in all cases the expectations.

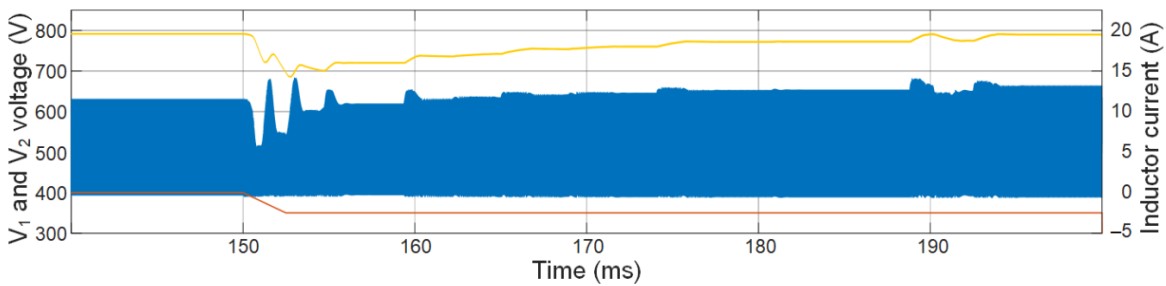

**Figure 5.** Simulation results of the dynamic behavior of the PI regulator after a step in the input voltage. $V_1$ changes from 400 V to 350 V in t = 150 ms. $V_1$ in brown, $V_2$ in yellow and $I_L$ in blue.

### 3.2. Improving Efficiency by Changing between Conduction Modes

Figures 6 and 7 summarizes the control strategy of the synchronous boost DC/DC converter based on selecting among conduction modes for different load levels. The efficiencies at high power levels for both conduction modes (i.e., QSW-ZVS and CCM-HS) are similar for the described specifications (estimated using the efficiency models and experimentally validated, Figure 3). This fact allows a possible digital control solution based not only on the efficiency but also on other factors, such as, peak current level or current ripple, which are advantageous strategies during the battery charging process (during the initial stage of charge, currents are considerably high). Other conditioning factor to decide about the selection of the conduction mode could be considered,

such as, electromagnetic interference (EMI) due to frequency variation, but it is out of the scope of this work.

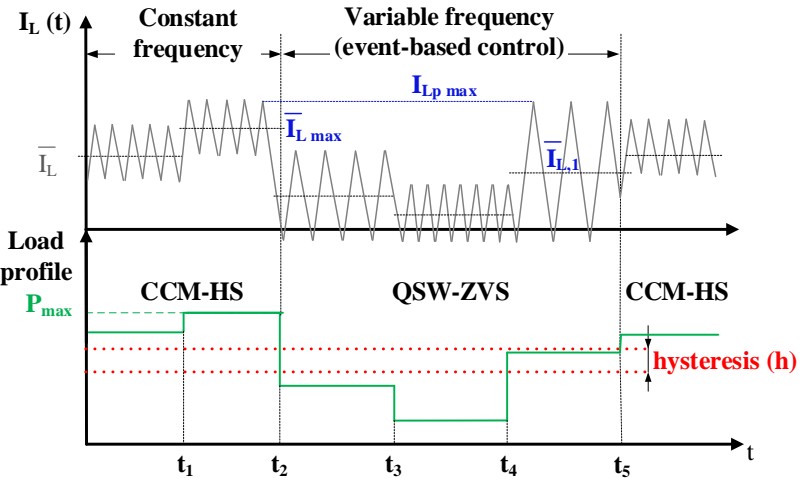

**Figure 6.** Example of switching among modes strategy depending on the load.

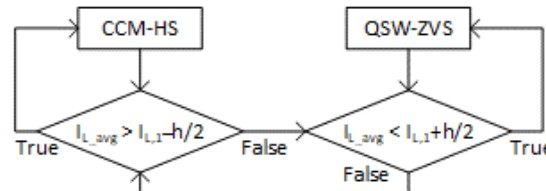

**Figure 7.** Flow chart explaining selection among modes.

Therefore, the algorithm of decision of the suggested digital control strategy attempts to reduce the current ripple and the peak current level through the semiconductors and inductor and, consequently, CCM-HS is preferred for high power levels. CCM-HS also provides easier control since it works at constant frequency. When the power falls and the peak current level of QSW-ZVS is similar to that of CCM-HS at full load (considering a certain hysteresis), QSW-ZVS is selected for low power to keep high efficiency. The average inductor current value associated to this peak current level is called $I_{L,1}$ in this paper (Figures 6 and 7).

Next, we are going to provide a numeric example with the prototype specifications in order to clarify this point. If the peak current value at 10 kW is 33 A operating in CCM-HS, the control strategy will change to QSW-ZVS when the power level is below 5.4 kW, which corresponds to 33 A of peak current value and 13.5 A average value for QSW-ZVS. In other words, $I_{L,1}$ equals 13.5 A. It is also important to note that below this power level the efficiency using QSW-ZVS is considerably higher than using CCM-HS.

This digital control solution and the estimation of the proper value of $I_{L,1}$ can be accomplished since all the data about the different conductions modes (switching frequencies, peak current values, efficiency, etc. for different loads) is given by the previously validated efficiency models [19] (introduced in Figure 3 results and developed in [19]). Moreover, the average current through the inductor ($I_{L\_avg}$) is experimentally measured using a high bandwidth current sensor twice every period at $DT/2$ and $((D + 1) T)/2$ (see Figure 8). Later, the average value of these two measurements is compared with the reference $I_{L,1}$ and a decision about the best conduction mode is taken every switching period. A hysteresis condition is considered in order to avoid loss of control for boundary situations (i.e., when $I_{L\_avg}$ is exactly $I_{L,1}$ any variation in load, noise or any other source of disturbance could make the control continually shift the switching frequency, Figure 6).

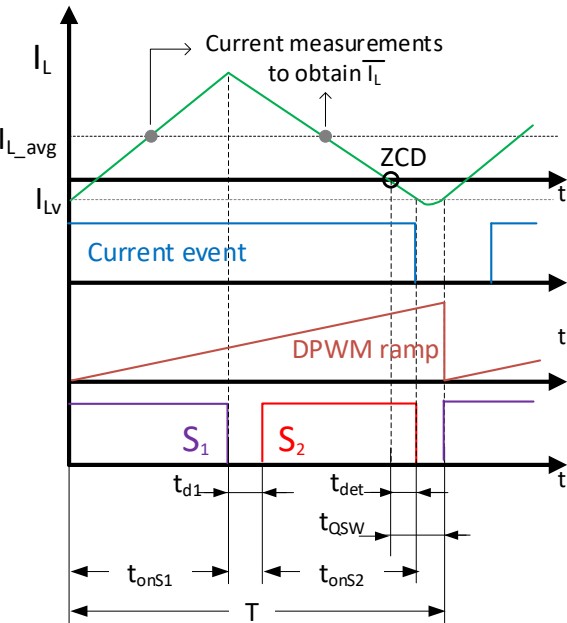

**Figure 8.** Main waveforms of the event-based DPWM with variable switching frequency implemented for QWS-ZVS operation: $I_L$ (green), current event detection signal (blue), DPWM ramp (brown) and gate signals ($S_1$ in violet, $S_2$ in red).

As a result, the Conduction mode selection block selects CCM-HS or QSW-ZVS depending on the value of $I_{L\_avg}$ given by the current sensor.

In Figure 9, simulation results for changes among modes are presented. In Figure 9a, the power goes from $V_1$ to $V_2$ (similar experimental results are presented in Section 4). In Figure 9b, the power goes from $V_2$ to $V_1$ (the control is automatically adapted) to validate the bidirectional operation.

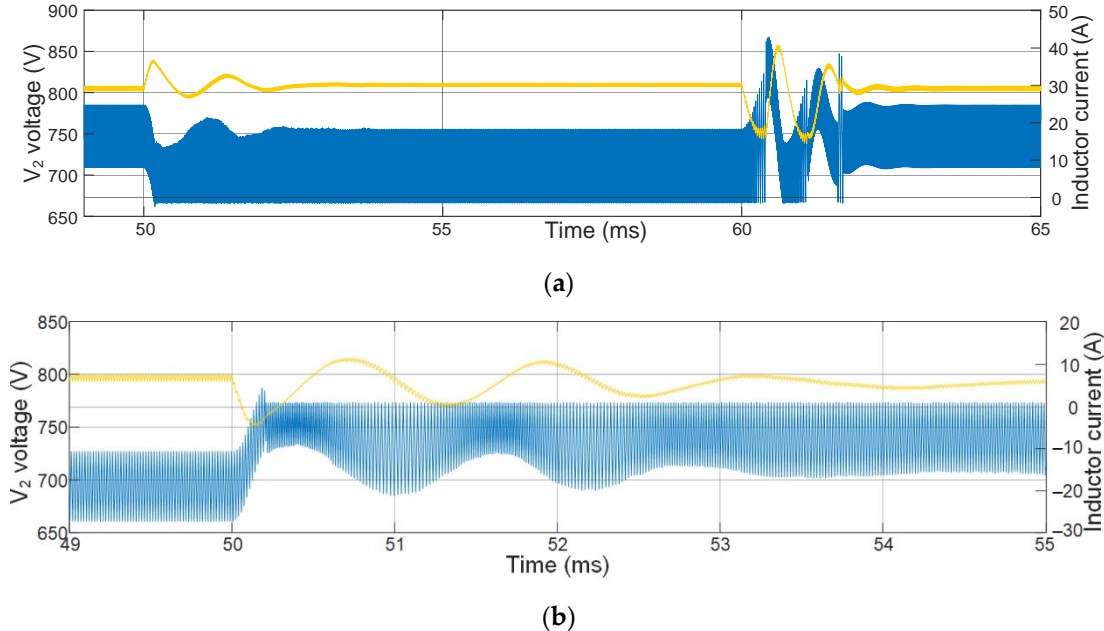

(**a**)

(**b**)

**Figure 9.** Simulation results changing among conduction modes. (**a**) Power flows from $V_1$ to $V_2$ and changes from 6.2 kW to 3.5 kW at t = 50 ms and from 3.5 kW to 6.2 kW at t = 60 ms are promoted. (**b**) Power flows from $V_2$ to $V_1$ and change from 8 kW to 3.2 kW at t = 50 ms is promoted. $V_2$ in yellow and $I_L$ in blue.

### 3.3. Event-Based DPWM with Variable Switching Frequency to Operate in QSW-ZVS

The DPWM signals at constant frequency for HS-CCM operation can be easily implemented. A digital ramp with a constant period is generated and the duty cycle given by the closed loop voltage regulator and constant dead-times are used to generate the gate signals. Nevertheless, to assure QSW-ZVS (meaning, variable switching frequency) a more attentive implementation of the digital control solution is required (Figure 8).

A switching period is chosen by the frequency selection block subjected to the conduction mode. There are no calculations needed when CCM-HS is selected, but the block requires extra-data from the current sensor block if QSW-ZVS is the selected conduction mode. In this paper, simple event detection is used, while in previously presented papers [22], more complex analytical estimations are required.

The interval of time where $S_1$ is conducting ($t_{onS1}$) is given by the voltage regulator block, to guarantee the desired voltage. However, $S_2$ conduction time ($t_{onS2}$) depends on Zero Current Detection (ZCD) to assure QSW-ZVS operation (Figure 8). It is hard to find a current sensor with a bandwidth big enough to obtain an accurate digital detection of the ZCD. For this reason, the output of the current sensor is analogically compared with a reference providing an event to the FPGA. This event is detected by the FPGA within a certain delay that remains constant for any switching frequency ($t_{det}$). The value of the DPWM ramp in the instant where the FPGA detects this current event is the used to determine the following switching period. This selected value is compared to its minimum and maximum value to prevent operating out of the switching frequency limits.

After the ZCD is detected, a certain time ($t_{QSW}$) is set, where the resonance takes place. This time is big enough to allow the energy stored in the inductance discharges the output parasitic capacitance of the SiC MOSFET achieving ZVS. The delay between ZCD and the current event does not compromise the accuracy of this method, since this time delay is constant for different switching frequencies and is lower than $t_{QSW}$. Finally, after the deadtime finishes, $S_1$ is turned on, starting a new period. Constant dead-times to avoid shoot-through, $t_{d1}$ and $t_{d2}$ (included in $t_{QSW}$) are used.

In Figure 10, simulation results showing changes of the switching frequency in QSW-ZVS for load steps are presented. As can be seen, the negative value of the inductor current is kept constant, while the switching frequency is modified to change the mean inductor current (similar experimental results are presented in Section 4).

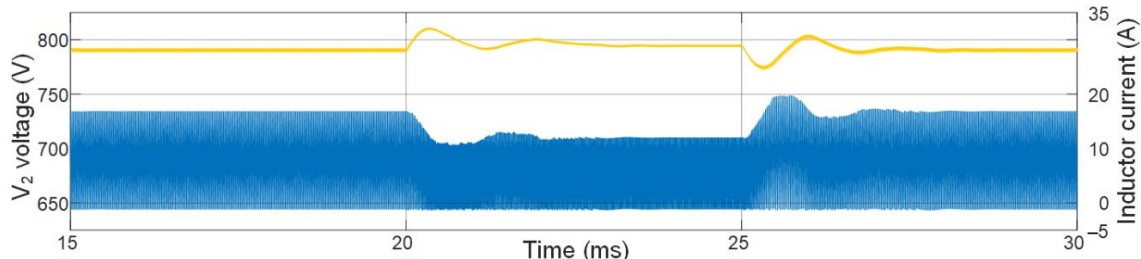

**Figure 10.** Simulation results for load steps in QSW-ZVS. Power flows from $V_1$ to $V_2$ and changes from 3 kW to 2 kW at t = 20 ms and from 2 kW to 3 kW at t = 25 ms are provided. $V_2$ in yellow and $I_L$ in blue.

The required switching frequency in QSW-ZVS for each power can be analytically or graphically estimated using [14]. However, the frequency is automatically achieved using the event-based strategy. In Figure 11, two simulation results examples show the accuracy of the proposed method. Slight mismatch is mainly due to differences of the minimum value of the $I_L$ given during the resonant period. A good agreement with experimental results is also obtained (see Figure 16c).

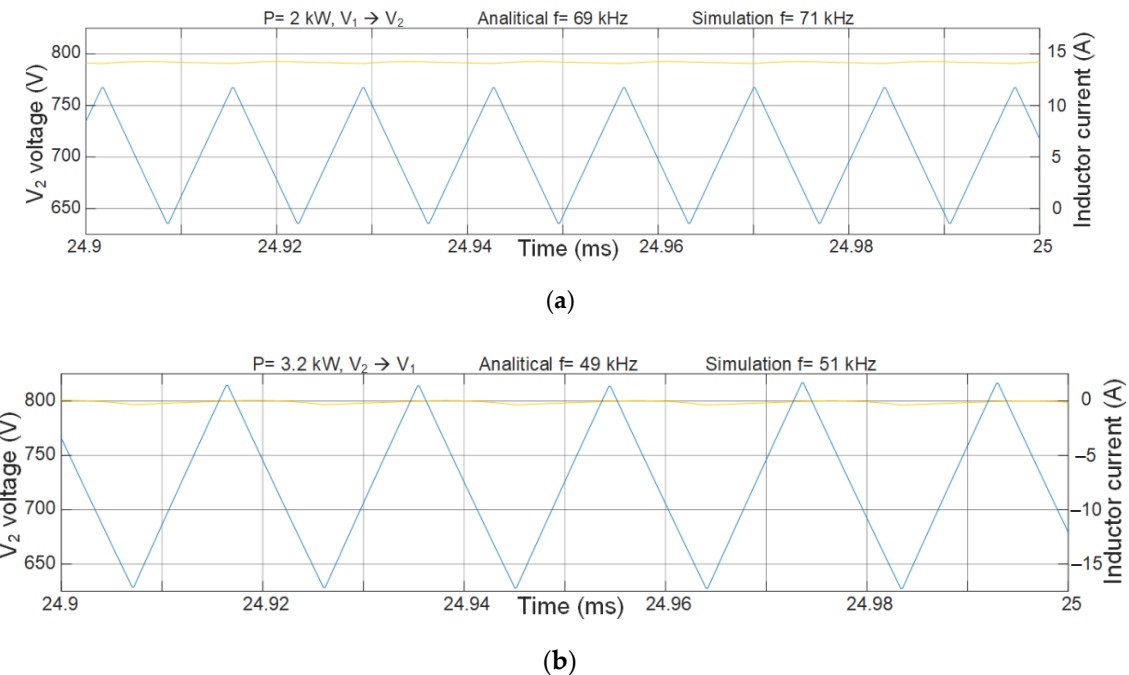

(a)

(b)

**Figure 11.** Simulation results showing the switching frequency on QSW-ZVS for different power. (**a**) $p$ = 2 kW flows from $V_1$ to $V_2$. (**b**) $p$ = 3.2 kW flows from $V_2$ to $V_1$. $V_2$ in yellow and $I_L$ in blue.

## 4. Experimental Results

A SiC-based synchronous boost DC/DC converter prototype has been developed to validate the proper operation of the proposed control strategy, which is composed of the blocks showed in Figure 4. The prototype (Figure 12) consists of the MOSFET module CCS020M12CM2 [23] from CREE® (three half-bridge six-pack module) and the 3-channel driver CGD15FB45P1. The value of the inductor is 200 μH (one inductor of 600 μH per branch, 3 branches in parallel). It should be noted that no modularization technique is applied in these experimental results and the three HB receive the same control signals. Therefore, they share the voltages, currents, and power. The FPGA under current use is Nexys 4 DDR based on a ARTIX 7 from XILINX ® (California, USA). Input and output capacitors of 4 μF and 12 μF are used, respectively.

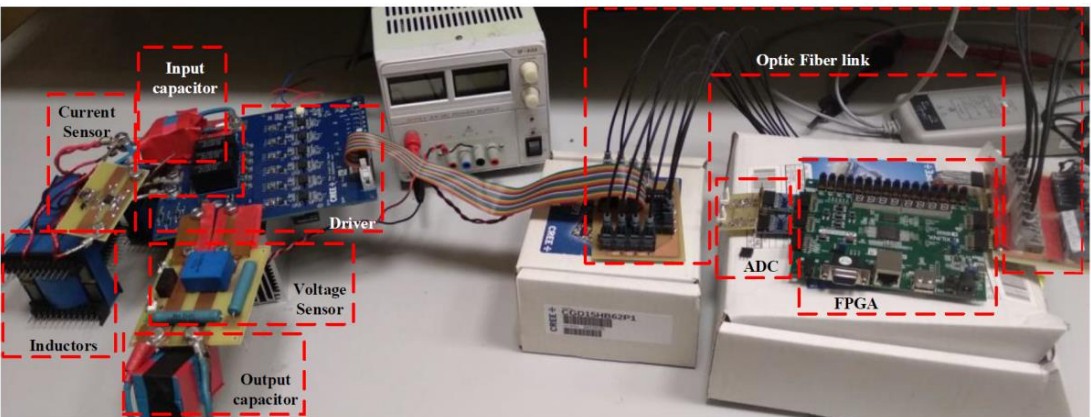

**Figure 12.** Prototype: power stage, current and voltage sensors on the left part of the image and optic fiber link and control stage on the right part of the picture.

Regarding the current sensor, it is especially interesting to show the current event detection in which this work is focused. To digitally detect the change of sign of the inductor current, an ADC and a current

sensor with high bandwidth (BW) would be required. It is easy to find ADCs with high BW, but the BW of commercially available current sensors is not enough for the switching frequencies used in this paper. Consequently, an analogical detection is implemented to obtain this event using a high BW current sensor (CQ3200 with BW = 500 kHz) and a high-speed comparator with internal hysteresis (LMV7219).

In Figure 13, a test in QSW-ZVS operation is shown. When the current through the inductor goes below 0 A, the current event signal has a falling edge. This event arrives to the control after a certain delay ($t_{det}$). At that moment, the control stores the time spent from the beginning of the switching period to that instant in order to stablish the next switching period. As can be seen, $t_{det}$ is shorter than the time needed to achieve ZVS ($t_{QSW}$). We define $t_{QSW}$ as the length of time during which the current goes negative and the output parasitic capacitance of the device is discharged.

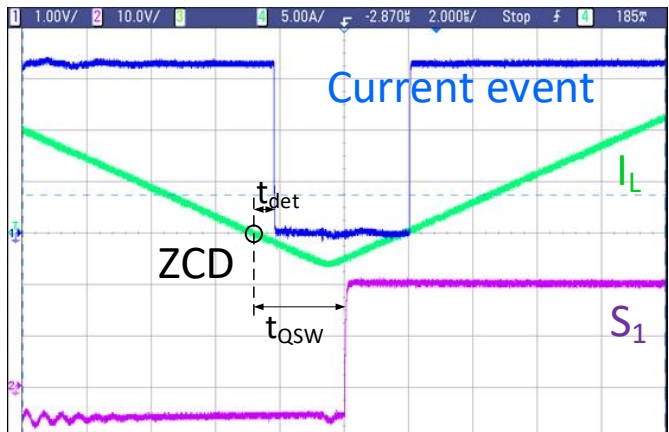

**Figure 13.** Current event detection based on the current sensor and ZCD. Gate signal $S_1$ (violet), $I_L$ (green) and current event signal (blue).

*4.1. Voltage Regulator*

In Figure 14, a step in $V_1$ is shown to check the dynamic response of the voltage regulator in close loop operation. As previously mentioned, the voltage regulator design is not a contribution of this paper, but it is needed to regulate $V_2$. Experimental results are shown to validate the smooth operation of the voltage regulator and the simulation results previously stated (the response time is around 40 ms, which matches with the results of Figure 5).

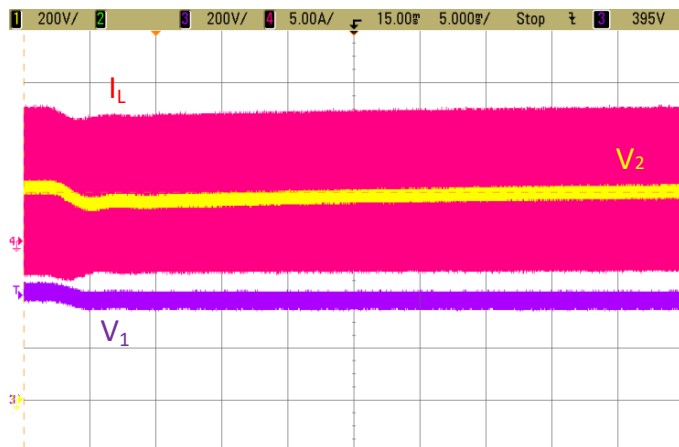

**Figure 14.** Dynamic behavior of the PI regulator over a step in the input voltage. $V_1$ changes from 400 V to 350 V, keeping $V_2$ = 800 V. $V_1$ (violet), $I_L$ (pink) and $V_2$ (yellow).

### 4.2. Changing among Conduction Modes

In Figure 15, a load step is introduced to validate the change among modes, which is the objective of our proposal to improve efficiency. The control is able to detect the average value of the inductor current, $I_{L\_avg}$, and compares it with a certain value, $I_{L,1}$ +/− h/2 and sets the conduction mode, and consequently the switching frequency. When the control decides that QSW-ZVS is preferred, it also establishes the switching frequency needed for the load.

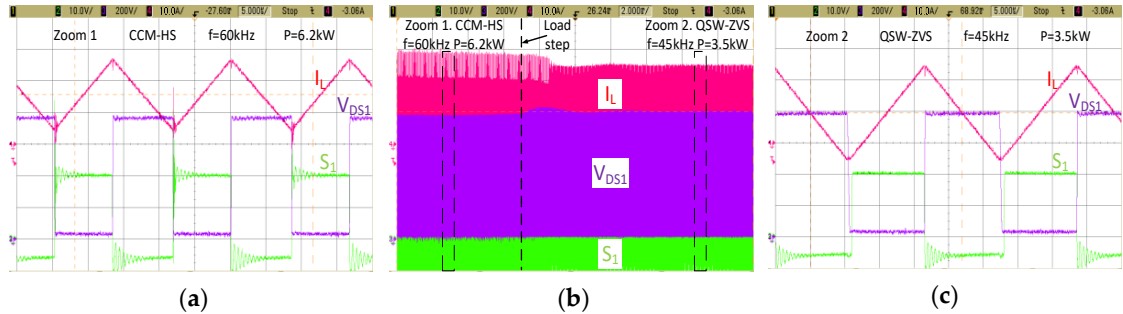

**Figure 15.** Experimental results showing the change from CCM-HS to QSW-ZVS after a load step (6.2 kW to 3.5 kW) by detecting $I_{Lavg} < I_{L,1}$. (**a**) Zoom of CCM-HS operation, (**b**) general view and (**c**) zoom of QSW-ZVS operation. $V_{DS1}$ (violet), $I_L$ (pink) and $S_1$ (green).

In Figure 15b, a general view of the load step performed is shown, with an appreciable load change going from 6.2 kW ($I_{L\_avg} \approx 15.5$ A) to 3.5 kW ($I_{L\_avg} \approx 9$ A). As it was explained in previous sections, the boundary average inductor current in this application is set for $I_{L,1} = 13$ A. When $I_{L\_avg}$ goes under/over this value the conduction mode that optimizes efficiency changes.

In Figure 15a, a zoom of operation before the load step is displayed. CCM-HS at a constant switching frequency of 60 kHz is selected. The current through the inductor is always positive and with low current ripple.

In Figure 15c, a zoom of operation after the load step is presented. In this case, QSW-ZVS with its adequate switching frequency of 45 kHz for the output power processed is chosen. It is noticeable how $V_{DS}$ goes to zero before the turn-on of switch $S_1$, providing ZVS. In this case, the minimum value of the inductor current must be negative for obtaining ZVS, and therefore, the current ripple is appreciable in QSW-ZVS operation.

In Figure 16, a load step is introduced to check how the control changes the switching frequency when operating in QSW-ZVS below the threshold $I_{L,1}$.

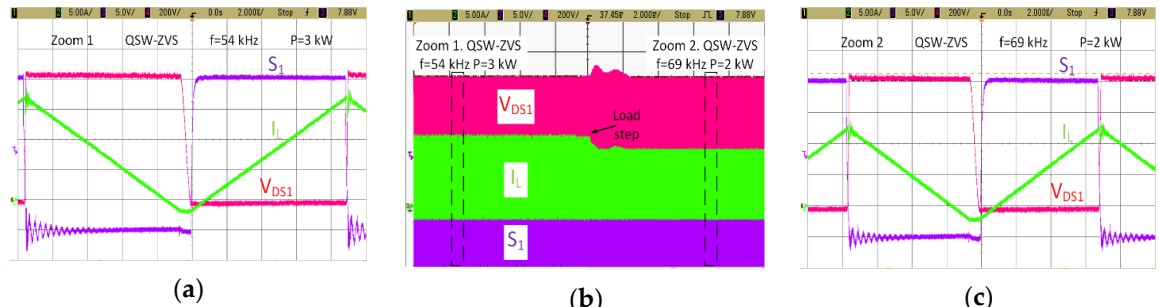

**Figure 16.** Experimental results showing the automatic change of the switching frequency in QSW-ZVS under a load step (3 kW to 2 kW). (**a**) Zoom at 3 kW and 54 kHz, (**b**) general view and (**c**) zoom at 2 kW and 69 kHz. $V_{DS1}$ (pink), $I_L$ (green) and $S_1$ (violet).

In Figure 16b, a general view of the load step from 3 kW to 2 kW is shown. Since both power demands correspond to inductor average currents lower than the threshold, the selected conduction

mode is QSW-ZVS, but the switching frequency must be different. In Figure 16a, a zoom of operation before the load step shows an automatically selected switching frequency of 54 kHz. In Figure 16c, a zoom after the load step shows a switching frequency of 69 kHz.

## 5. Conclusions

The use of a digital control in a SiC-based synchronous boost DC/DC converter allows for the design of a control strategy based on selecting among different conduction modes (CCM-HS and QSW-ZVS) depending on the load demand to keep the efficiency almost constant in a wide operation range and to reduce peak current levels at high output power, which is beneficial for certain applications, (e.g., battery chargers).

An event detection approach is proposed to develop DPWM at variable switching frequency, assuring QSW-ZVS operation for different loads.

The proposed control strategy, integrating changing among conduction modes and the event-based DPWM at variable switching frequency is developed in an FPGA and validated in a SiC-based synchronous boost DC/DC converter. Moreover, simulation results are presented to validate the bidirectional operation of the proposal.

High efficiency (99–97%) over a wide power range (from 100% to 5% of the load) and reduction of the peak current at full load are obtained with the presented strategy.

**Author Contributions:** Conceptualization, A.R.; methodology, D.G.L.; software, M.R.R.; validation, A.R., and M.R.R.; formal analysis, A.V.; resources, M.M.H.; writing—original draft preparation, M.R.R.; writing—review and editing, A.R. and D.G.L.; supervision, A.V.; project administration, M.M.H. All authors have read and agreed to the published version of the manuscript.

**Funding:** This work was supported in part by the Spanish Government under Project MCIU-19-RTI2018-099682-A-I00 and by the Principality of Asturias under Project FC-GRUPIN-IDI/2018/000179.

**Conflicts of Interest:** The authors declare no conflict of interest.

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
