# Peer review of "Event-Focused Digital Control to Keep High Efficiency in a Wide Power Range in a SiC-Based Synchronous DC/DC Boost Converter"

_electronics, doi:10.3390/electronics9122154_

Round 1

Reviewer 1 Report

This is a good work overall. I suggest to add a section to compare with prior work and highlight the contribution of this paper.

There is couple things need to be clarified. 

In Fig 3. there is negative power in the plot. If the negative sign for the power is for the direction of the conversion, it needs to state which direction is the positive. If it has other meanings, it should state clearly what the sign of the power represents.

In Fig 4. There is no port V1, but V1 is referred in the line 109. 

In Fig 7. the flow chart looks wrong for the sign of hysteresis. For example, if I L_avg = I L,1, there will be dead-loop in the logic.

Author Response

Dear Editors and Reviewers:

First of all, we are grateful for the review comments and the opportunity of revision. The authors have seriously analyzed and discussed the questions raised by the reviewers, and all the comments are addressed in the revised manuscript. According to the comments, the manuscript is mainly revised as follows:

  • Some contents are complemented for an easier comprehension. Changes in the revised manuscript are marked in red.
  • Other contents are re-structured. These changes are marked in green in the revised manuscript.

A general critic made the reviewers was about the main contribution, objective and purpose of this work. Please, find that the Introduction Section has been re-structured in order to fulfil this requirement.

And our replies to the reviewers are presented in the following.

Reviewer 1

We wish to thank the reviewer for her/his helpful comments. The reviewer has made a few comments (in black color), and our responses are:

This is a good work overall. I suggest to add a section to compare with prior work and highlight the contribution of this paper.

Authors have highlighted the objectives and main contributions of this work by re-stucturing the Introduction Section.

There is couple things need to be clarified. 

In Fig 3. there is negative power in the plot. If the negative sign for the power is for the direction of the conversion, it needs to state which direction is the positive. If it has other meanings, it should state clearly what the sign of the power represents.

The reviewer is completely right. Authors have clarified this aspect in Figure 3 caption. Positive power is considered when the power flows from V1 to V2. And, conversely, negative power represents power flowing from V2 to V1.

In Fig 4. There is no port V1, but V1 is referred in the line 109. 

It is true that V1 does not appear in Fig. 4. This is due to the fact that the control proposed does not depend directly on V1 voltage value. Instead, the control (specific for a boost behavior) assures a certain predefined V2 voltage value. For the sake of simplicity, when V1 is named in the text, Fig. 1 is referred.

In Fig 7. the flow chart looks wrong for the sign of hysteresis. For example, if I L_avg = I L,1, there will be dead-loop in the logic.

The reviewer is right. There is an error in the flow chart. The signs of the hysteresis under comparison are swapped. Authors have modified the signs accordingly.

Reviewer 2 Report

The Authors essentially proposed a digital control using a method using DPWM event detection to switch to modes of conduction of QWS-ZVS and CCM-HS to increase the efficiency of the SiC-based synchronous boost dc/dc converter. I think overall, the presentation of the paper is poor and needs extensive editing. Moreover, I have the following comments:

  • Why the authors select SiC-based switches for the dc-dc converter, is their proposed approach not working for the Si-based converters? What essential characteristics or drawback of SiC-based converters that the proposed control strategy can address? We know that due to load variation, SiC-based converters may show significant ringing/overshoot is transient due to parasitic elements how the control methodology will handle the transients. 
  • I am wondering what significant improvement this paper suggests compared to the author's paper in the APEC conference (Ref [19])? In the list of references, the authors have five self-citations [13, 14, 18, 19, 20]. I believe this level of self-citation is, first of all, excessive, and the authors have not justified the real contribution of this manuscript in comparison to their previous work. The clarification of the contribution is necessary. 
  • The figures in Sections 3 and 4 are not presented well. I believe all the figures for simulation and experiments have to compared side-by-side for the reader to see the agreement, and improvement in practice quickly. 
  • The authors mentioned in Page 10, "As it was explained in the previous 280 sections, the boundary average inductor current in this application is set for IL,1 =13 A." Why the authors select the IL,1 =13 A? Should this boundary condition be automatically selected in the control algorithm based on the load conditions? The authors mentioned in page 6, "A hysteresis condition is contemplated to avoid loss of control for boundary situations (i.e. when IL_avg is exactly IL,1 any variation in load, noise or any other source of disturbance could make the control continually shift the switching frequency)." Can the authors clarify how they exert hysteresis for determining boundary conditions? An illustrative figure would be useful.
  • The paper needs extensive editing in English, for instance, on page 10, "It is appreciable 287 how VDS goes to zero previously the turn-on of switch S1..." This is bad English. The paper needs extensive editing for a smooth readout.  

Author Response

Dear Editors and Reviewers:

First of all, we are grateful for the review comments and the opportunity of revision. The authors have seriously analyzed and discussed the questions raised by the reviewers, and all the comments are addressed in the revised manuscript. According to the comments, the manuscript is mainly revised as follows:

  • Some contents are complemented for an easier comprehension. Changes in the revised manuscript are marked in red.
  • Other contents are re-structured. These changes are marked in green in the revised manuscript.

A general critic made the reviewers was about the main contribution, objective and purpose of this work. Please, find that the Introduction Section has been re-structured in order to fulfil this requirement.

And our replies to the reviewers are presented in the following.

Reviewer 2

We wish to thank the reviewer for her/his helpful comments. The reviewer has made a few comments (in black color), and our responses are:

Why the authors select SiC-based switches for the dc-dc converter, is their proposed approach not working for the Si-based converters? What essential characteristics or drawback of SiC-based converters that the proposed control strategy can address? We know that due to load variation, SiC-based converters may show significant ringing/overshoot is transient due to parasitic elements how the control methodology will handle the transients.

The reviewer has a point questioning the suitability of the proposed control for Si based converters. Probably, generally speaking, the strategy proposed could be aplicable in converters with Si switches (although it should be experimentally validated). However, there are several reasons why the authors chose a SiC based dc-dc converter.

In the first place, the application for which this solution was proposed is a 10kW converter, connecting a 400V storage system with a 800V cell of a multilevel converter. For this range of voltage, at least 1200V devices should be considered (taking into account possible over-voltages). In addition, high power density is searched (and for that, high-switching frequency is preferred), while keeping high efficiency for the whole load range. It is well known that in this sense SiC technology offers good performance under high requiring working conditions.

In the second place, regarding transients, the proposed control is able to achieve Zero Voltage Switching, which reduces importantly ringing/overshoots. Moreover, this solution is thought for applications where storage systems are needed and not too big load steps are expected, reason why the voltage regulator does not need to be too fast.

Authors have re-structured the Introduction Section and have made emphasis in the applicability of the solution proposed.

I am wondering what significant improvement this paper suggests compared to the author's paper in the APEC conference (Ref [19])? In the list of references, the authors have five self-citations [13, 14, 18, 19, 20]. I believe this level of self-citation is, first of all, excessive, and the authors have not justified the real contribution of this manuscript in comparison to their previous work. The clarification of the contribution is necessary. 

Certainly, the present manuscript is a further detailed version of previous conference papers [19-20] where more insight of the design and analytical validity of the control solution proposed is provided. Also, it takes into account some concepts deeper defined in [13-14,18]. If authors do not cite previous work, comprehension of some of the states of this manuscript would be more difficult, and it will lead to a longer and repetitive manuscript.

The main contribution of this work resides on a deeper analysis of the control, where simulation models and results are presented. Besides, the voltage regulator and the procedure to change between conduction modes based on a even-focused strategy is validated.

Authors have re-structured the Introduction Section and as stated before, they have clarified the main contribution of the manuscript.

The figures in Sections 3 and 4 are not presented well. I believe all the figures for simulation and experiments have to compared side-by-side for the reader to see the agreement, and improvement in practice quickly. 

Several text structures may be applicable for a good comprehension of the manuscript. However, authors have decided to keep the same organisation, differentiating simulation from experimental results. In any case, it would be difficult to have all graphs side-by-side to compare them at a glance.

The authors mentioned in Page 10, "As it was explained in the previous 280 sections, the boundary average inductor current in this application is set for IL,1 =13 A." Why the authors select the IL,1 =13 A? Should this boundary condition be automatically selected in the control algorithm based on the load conditions? The authors mentioned in page 6, "A hysteresis condition is contemplated to avoid loss of control for boundary situations (i.e. when IL_avg is exactly IL,1 any variation in load, noise or any other source of disturbance could make the control continually shift the switching frequency)." Can the authors clarify how they exert hysteresis for determining boundary conditions? An illustrative figure would be useful.

To undertand why authors have selected IL,1=13A, the reviewer should refer to Figure 6, where the hysteresis concept is reflected. This boundary value is chosen because for the power corresponding to this average current value and working under QSW-ZVS operation mode, a peak current equal to ILpmax is reached (being ILpmax the maximum peak current obtained at maximum power and under CCM operation mode).

The paper needs extensive editing in English, for instance, on page 10, "It is appreciable 287 how VDS goes to zero previously the turn-on of switch S1..." This is bad English. The paper needs extensive editing for a smooth readout. 

Line 287 has been modified and Authors have revised the manuscript trying to make a better use of english.

Reviewer 3 Report

The manuscript electronics-1001483 reports the design of a control strategy, based on events and changing among conduction modes.
The described strategy allows automatic modifications of the conduction mode and the optimization of the efficiency for different power ranges. Presented strategy, the Authors have validated with use a SiC-based synchronous boost DC/DC converter rated for 400 V to 800 V and 10 kW. This manuscript presented both: simulations with the use of Matlab-Simulink environment and experimental results.
The presented analysis and results are interesting, but there are a few points which should be clarified before I can recommend publication:

1) Figure 1 is almost identical to the figure in [13] - different letter symbols. If authors use figures from other papers, this should be clearly stated.
2) Figure 2 - The Authors should add references.
3) In conclusion, the authors should briefly compare the proposed strategy and previous similar solutions.
4) In my opinion, maybe in the summary, the Authors should present application proposals of their solution.
5) Line 28 is: "transformers [1]-[6]." In my opinion, it will be better [1-6] - see also another part of the paper.
6) Line 46 is: "and high frequency [13], [14]." In my opinion, it will be better [13,14] - see also another part of the paper.
7) Lines 56-63: Sections are numbered by Roman numerals. Authors should use Arabic numbers.
8) Figures 12 and 13 should be in the reverse order. In the manuscript, fig. 13 is described first, and then fig. 12.
9) Reference 13 - no year of publication.

Author Response

Dear Editors and Reviewers:

First of all, we are grateful for the review comments and the opportunity of revision. The authors have seriously analyzed and discussed the questions raised by the reviewers, and all the comments are addressed in the revised manuscript. According to the comments, the manuscript is mainly revised as follows:

  • Some contents are complemented for an easier comprehension. Changes in the revised manuscript are marked in red.
  • Other contents are re-structured. These changes are marked in green in the revised manuscript.

A general critic made the reviewers was about the main contribution, objective and purpose of this work. Please, find that the Introduction Section has been re-structured in order to fulfil this requirement.

And our replies to the reviewers are presented in the following.

Reviewer 3

We wish to thank the reviewer for her/his helpful comments. The reviewer has made a few comments (in black color), and our responses are:

1) Figure 1 is almost identical to the figure in [13] - different letter symbols. If authors use figures from other papers, this should be clearly stated. 2) Figure 2 - The Authors should add references.

The reviewer is right. Now, Figures 1 and 2 are referred to previous work[13].

3) In conclusion, the authors should briefly compare the proposed strategy and previous similar solutions. 4) In my opinion, maybe in the summary, the Authors should present application proposals of their solution.

Typical applications for the solution proposed could be wind energy generation with storage capability or electric vehicle battery chargers.

The main contribution of this work resides on a deeper analysis of the control, where simulation models and results are presented. Besides, the voltage regulator and the procedure to change between conduction modes based on a even-focused strategy is validated.

Authors have re-structured the Introduction Section and have made emphasis in the applicability of the solution proposed and which is the main contribution of this manuscript.

5) Line 28 is: "transformers [1]-[6]." In my opinion, it will be better [1-6] - see also another part of the paper. 6) Line 46 is: "and high frequency [13], [14]." In my opinion, it will be better [13,14] - see also another part of the paper.

Citations have been modified to meet with the requirements for publishing.

7) Lines 56-63: Sections are numbered by Roman numerals. Authors should use Arabic numbers.

The reviewer is right. Sections have been renamed with arabic numerals.

8) Figures 12 and 13 should be in the reverse order. In the manuscript, fig. 13 is described first, and then fig. 12.

Figures 12 and 13 have been swapped to follow the order in which they are described in the text.

9) Reference 13 - no year of publication.

Vazquez, A. Rodriguez, M. R. Rogina and D. G. Lamar, "Different Modular Techniques Applied in a Synchronous Boost Converter With SiC MOSFETs to Obtain High Efficiency at Light Load and Low Current Ripple," in IEEE Transactions on Industrial Electronics, vol. 64, no. 10, pp. 8373-8382, Oct. 2017.

Reference [13] has been updated.